# Cold spray as a powder metallurgy process for production of nickel aluminium bronze

**Steven Camilleri**[1,2]**, Tien Tran**[1,2]**, Andrew Duguid**[2]**, Kannoorpatti Narayanan Krishnan**[1]*

**1** Advanced Manufacturing Alliance, Energy and Resources Institute, Charles Darwin University, Darwin, Australia, **2** SPEE3D, Charles Darwin University, Casuarina NT, Australia

* Krishnan.kannoorpatti@cdu.edu.au

## Abstract

Nickel aluminium bronze (NAB) alloys are known for their excellent strength and corrosion resistance, making them suitable for maritime and industrial applications. NAB is producible by Powder Metallurgy (PM) but typically requires high compaction pressure. The objective of this study is to investigate the manufacturing of NAB using the cold spray additive manufacturing (AM) process and to compare its properties to those produced by traditional methods such as casting and PM. Cold spray is a solid-state coating technique that accelerates powdered metal and carrier gas to supersonic speeds, enabling bonding through plastic deformation. Binary aluminium bronze (AB) and NAB alloys were produced using powders by cold spraying powders into 3D printed parts, and heat treating the resulting parts. The AB alloy contained blended 9.9% aluminium alloy (Al6061) powder and copper powder, while the NAB alloy included 11% Al6061 powder, 5.8% nickel powder, 6.8% iron powder, and copper powder. Powders were mixed under controlled conditions and deposited using a LightSPEE3D printer and compressed air. Post-deposition heat treatments, such as homogenisation, aging, and/or hot isostatic pressing (HIP), were applied to enhance material properties. The results indicate that the cold spray process, combined with appropriate heat treatments, can produce NAB alloys with desirable microstructures containing fine κ phases and mechanical properties with above 280 MPa yield strength, above 500 MPa tensile strength and 20% elongation which are comparable to those achieved by traditional cast methods which yield strength of 240 MPa, tensile strength of 580 MPa and 15% elongation, and superior to PM methods. This study demonstrates the viability of cold spray AM to enhance the production of complex high-strength alloys, offering significant advancements for maritime and industrial applications.

## Introduction

Metal manufacturing has evolved significantly over the centuries, transitioning from traditional methods like casting and forging to more sophisticated techniques such as sheet metal forming and powder metallurgy (PM). In recent decades, additive manufacturing (AM) of metal has emerged, becoming increasingly commercially viable and reliable, particularly over the last ten years.

**Data availability statement:** All relevant data are within the article and its Supporting Information files.

**Funding:** This research was supported by the Department of Education Australia through the Department of Education's National Industry PhD Program (NIPhD 35017). The views expressed herein are those of the authors and are not necessarily those of the Australian Government or the Department of Education. There was no additional external funding received for this study.

**Competing interests:** I have read the journal's policy and the authors of this manuscript have the following competing interests: Steven Camilleri, Tien Tran and Andrew Duguid are employees of SPEE3D. This has been declared in the authorship details.

Among these innovative methods, the cold spray technique, also known as gas dynamic cold spray, stands out as a uniquely solid-state coating process that propels powdered metal and a carrier gas to supersonic speeds through a de Laval nozzle. Unlike traditional thermal spray techniques, which often lead to significant thermal degradation, oxidation, or phase transformations, cold spray ensures that upon impact with a substrate, the particles undergo plastic deformation and bond to the surface or prior particles without significant heating. While AM is lauded for its ability to create objects with customisable geometries without the need for traditional tooling, it is less recognized for its ability to customise material properties. By mixing elemental metal powders rather than using prealloyed powders, various alloys can be created providing greater flexibility. This flexibility leads us to compare cold spray additive manufacturing with powder metallurgy. In this paper, the production of nickel aluminium bronze using cold spray method is compared with powder metallurgy process.

Nickel aluminium bronzes (NAB) are a popular set of alloys due to their excellent combination of properties. They have high strength and toughness, high wear, galling and cavitation resistances, are non-sparking, good corrosion and stress corrosion properties and biofouling resistance [1]. They are used extensively in aircraft landing gear bearings, marine propellers, pumps and valves, gears, and in non-sparking tools. They can be produced in cast, wrought or forged forms as well as by powder metallurgy. There are many different compositions of NAB alloy, but they generally contain up to 12% aluminium, 5% nickel, 7% iron, 3% manganese, and 2% silicon. The guide by Copper Development Association [1] is an excellent source of information on NAB. Aluminium is the primary element that strengthens the NAB alloys. Nickel and iron are added to improve corrosion and erosion resistance. Addition of iron also refines the structure and improves toughness. Manganese is added to improve fluidity of the alloys when casting.

When nickel and iron, about 5%, are added to copper-aluminium alloy, two phases α and κ are produced. When NAB is quenched from 900 °C, both α phase, martensite β and κ phases can form. On tempering at 500–715 °C, the martensite phase transforms to α (Cu-rich) and eutectoid phase (κ and $\gamma_2$) and secondary κ phases. The κ phase is found in 4 morphologies, locations and distributions. They are: $K_I$ – rosette form and iron-rich, $\kappa_{II}$ – spheroidised form at the grain boundaries, $\kappa_{III}$ – lath – shaped lamellar (Ni-rich) and $\kappa_{IV}$ – fine precipitate inside grains and iron-rich. The precipitates being rich in certain elements tend to get selectively corroded [1] as these phases set up galvanic couples with the other phases.

The corrosion resistance of NAB has been attributed to the passive layers of oxides of aluminium and copper [2]. The passive layers also provide excellent resistance to flow induced corrosion due to the tenacity of the protective films.

Various kappa phases containing Ni and Fe are known to increase the mechanical properties of NAB. A decrease in elongation has been attributed to excess κ phases which form with an increase in aluminium content above nominal.

Iron more than 3 wt% reduces grain growth at high temperatures and produces a smaller grain size and solidification range. Addition of iron allows NAB to retain strength at high temperatures [3]. Nickel addition refines grains by retarding β phase formation. However, nickel content should not be lower than the iron content as this will lower corrosion resistance. A nominal content of 5 wt % Ni and 4 wt % Fe provides the optimal high strength along with corrosion resistance [3]. The properties of NAB are dependent on how well the phases are refined and distributed in the alloy. If the κ phases in which segregation of elements occurs can be reduced, then the properties will be more uniform [4].

This NAB alloy are produced mainly by casting processes. This can also be produced by PM processes. These can be fabricated using welding processes. Recently, the NAB alloys are

being manufactured by AM methods. Different manufacturing processes produce different microstructures and properties based on processing history. Selected Laser Melting (SLM) fabricated NAB alloys produced uniform martensitic structures with nano-sized Fe-rich particles ($\kappa_{IV}$) [5]. Annealing transformed the microstructure into α and lamellar/spheroidized Ni-rich particles. Martensite and $\kappa_{IV}$ particles were not fully transformed.

Many of the Wire Arc Additive Manufacturing (WAAM) processes described in previous work significantly alter the microstructure of NAB producing grain growth and martensite in the heat affected zone [6–8]. The fusion zone exhibited Widmanstätten α and very fine martensite [9]. Martensite is harder and brittle and susceptible to corrosion and cavitation damage. After a Post-Weld Heat Treatment (PWHT) at 675 °C for six hours refined the martensite to α and κ phases. The WAAM parts generally had better or equal mechanical properties to that of cast materials. $\kappa_I$ was not precipitated due to the rapid cooling rates [7,9].

In additive manufacturing using beam processes, the powders used are generally alloy powders [8,10]. With the use of alloy powders, there is no flexibility to vary or compensate the composition of the alloy to get desired properties. Similarly, using WAAM, the wires used are standard wires and to make changes to the composition would require a whole new wire manufacturing run. Even powder metallurgy NAB products are manufactured using alloy powders [11,12]. The lack of compositional flexibility can be addressed by using other processes that can produce the required NAB alloy from elemental powders. Chen et al [13] had provided a review of the use of elemental powders for laser powder directed energy deposition process. Some of the issues highlighted are that each elemental powder will reflect laser beam differently and their addition into the melt pool is inconsistent. This would require remelting of each layer a second time. In some cases, some of the elemental powders failed to melt and/or evaporated at different rates, leading to compositional error in the final alloy.

Cold spray AM is an emerging technique in AM that provides advantages for material deposition and alloy production. Cold spray AM operates as a solid-state process, using pressurised gas to accelerate metal powders at supersonic speeds onto a substrate. This allows the formation of a cohesive and dense layer without melting. Elemental powders can be blended and cold spayed and heat treated to produce complex alloys instead of using pre-alloyed powder. Cold spray AM is a suitable manufacturing method for powder with widely varying melting temperature, making it ideal for producing complex alloys with customised properties [14].

The use of elemental powders in cold spray AM provides opportunities for improving the production process and the properties of NAB alloys. This is because the process does not melt the material, so vastly differing melting temperatures of the elements do not present a concern. Additionally, the elemental powders are a lower cost than alloyed powders. Production of NAB alloys by cold spray AM has been attempted by Peng et al using NAB prealloyed powder [16]. It was found that the deposition efficiency was only 7.61% and it increased to 30.56% after annealing at the powder at 600 °C for 10 hours. Heat treatment was done on the alloyed powder to modify the microstructure and to reduce the hardness. The deformation of hard, segregated alloy powder is less practical by cold spray unless a more expensive process relying on high pressure equipment or helium gas is used [17]. Prasad et al. demonstrated the repeatability and reproducibility of liquid-phase sintered aluminum bronze parts produced via cold spray additive manufacturing, with elongation increasing by up to 10%, and tensile strength improving by 2 to 4 times [15].

In this research, an attempt has been made to produce NAB which addresses some of the difficulties faced by other processes by using elemental powder, high pressure cold spray AM method and using air as the propellant.

## Materials and methodology

### Materials feed stock

Aluminium bronze was selected for cold spray AM to take advantages of using elemental blended with widely different melting temperatures. A binary aluminium bronze (AB) was fabricated by using cold spray process with a mixed powder of elemental copper and aluminium alloy (Al6061). Separately a nickel aluminium bronze (NAB) alloy was fabricated by using cold spray process with a mixed powder comprised of elemental copper, aluminium alloy (Al6061), elemental nickel and elemental iron powders. The composition ratios for AB were 9.9 wt% aluminium alloy Al6061 and the remainder copper. For NAB, the ratios were 11 wt% aluminium alloy, 5.8 wt% nickel, 6.8 wt% iron, and the remainder copper. The powders used in this process were gas-atomised powder and were supplied by SPEE3D.

The powders were mixed by a V-blender powder mixing machine for 1.5 hours at room temperature. Argon gas was introduced during mixing to displace the air inside the chamber and reduce the formation of oxide during mixing. Powder moisture was controlled by adding silica gel desiccant into the powder mixer for further removing moisture.

### Cold spray AM process

To form a cold spray compact, it is necessary to load the blended powder into the powder feeder of a cold spray AM apparatus, select a toolpath that will provide the right size and shape tensile specimen, set the cold spray process parameters, and then form the block. The cold spray AM apparatus chosen was a LightSPEE3D printer from the Australian company SPEE3D, equipped with their newer Phaser cold spray nozzle. The process gas was chosen to be compressed air in order to minimise process cost. The machine holds the cold-spray nozzle in a fixed position while a robot arm moves the workpiece over the particle-laden jet according to a deposition strategy determined by a proprietary 3D slicing algorithm. The carrier gas, air, is compressed to 30 bar, heated to 540 °C with 60 g/min. The standoff distance from the nozzle head to the build surface was fixed at 20 mm. Printed parts for tensile testing were rectangle blocks with dimensions of 20 mm x 32 mm x 106 mm in size. After forming the parts were subjected to different heat treatments and machined into groups of 5 tensile samples using EDM wire cutting (Fig 1) according to ASTM E8.

### Heat treatment process and testing

Press and sinter powder metallurgy relies on a wide range of sintering heat treatments to properly process a wide range of material systems. Advanced production PM furnaces are able to heat treat in hydrogen, or disassociated ammonia, or even inert atmospheres. In our case the sintering was carried out using an atmospheric furnace heat treatment (i.e., with minimum sintering cost) and in a separate experiment, using hot isostatic pressing using argon gas. This was done so a determination could readily be made if it was possible to obtain standard properties for the material system with casting.

The as-sprayed AB parts were homogenised at 1000 °C for 14 hours and furnace cool to 900 °C for 2.5 hours and quench in soluble oil. The as-sprayed NAB parts were homogenised at 1000 °C for 14 hrs, increase temperature to 1035 °C for 2.5 hours and furnace cool to 900 °C for 1 hours and quench in soluble oil then followed by aging at 625 °C for 2 hrs followed by quenching in soluble oil (Fig 2). With inclusion of HIP (Hot Isotactic Pressing) process, after the homogenisation step at 1000 °C, the parts were then HIPed at 1035 °C and 1000 bar for a 5-hour dwell, followed by a solution treatment at 1000 °C to dissolve precipitates caused

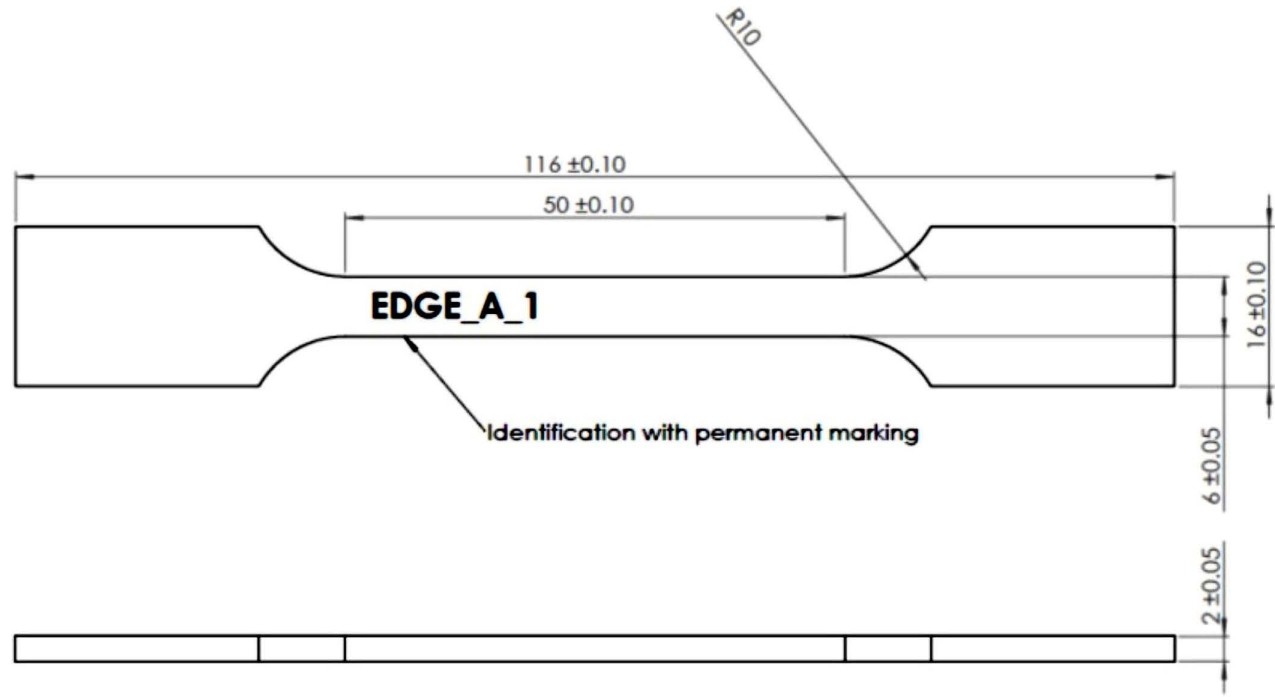

**Fig 1. Tensile sample design.**

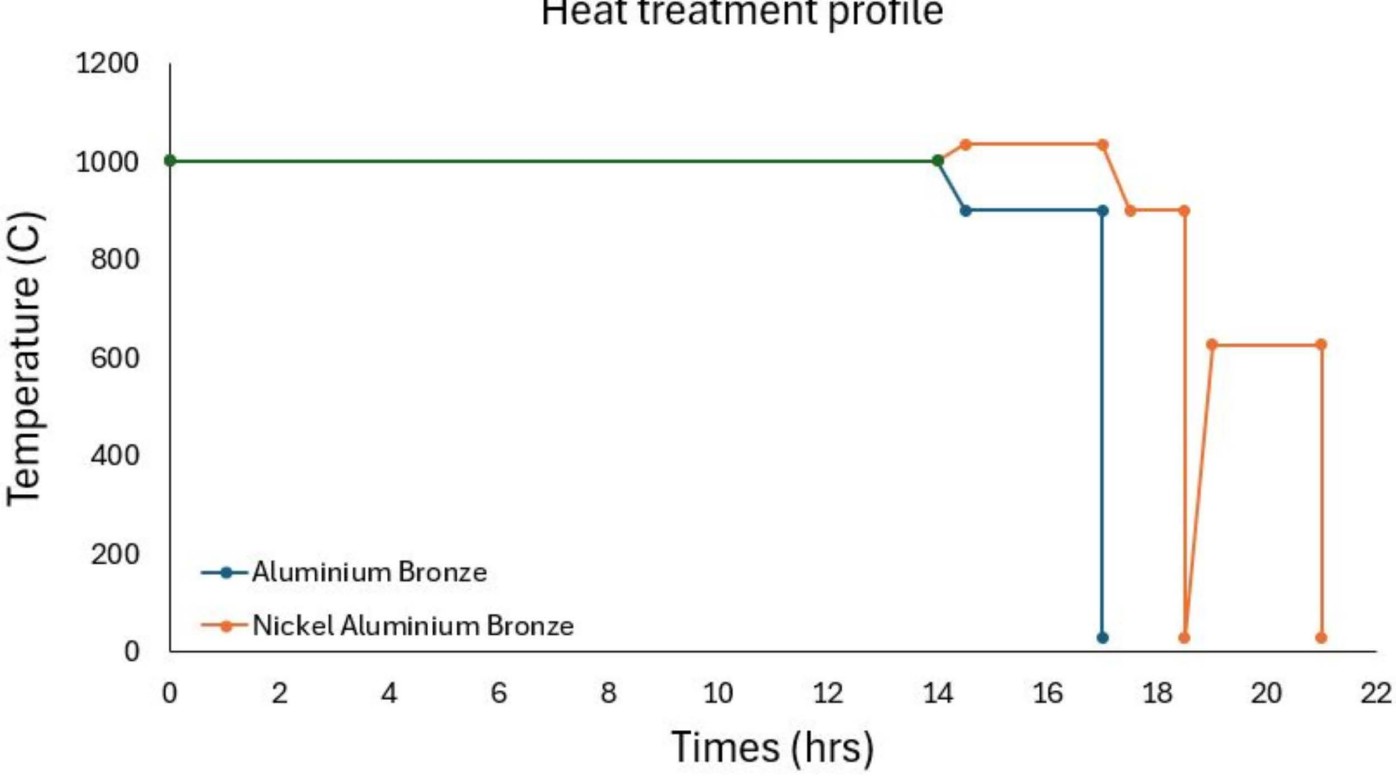

**Fig 2. Heat treatment profile for each material.**

by the slow cooling process after HIPing. The parts were then aged at 625 °C for 2hrs. The temperatures were chosen to produce sintering and homogeneous alloy based on previous literatures on PM on a similar alloy [18].

## Metallography and X-ray diffraction

The specimens were polished to 0.6µm finish and etched with Carpenter etchant (FeCl3, CuCl2, Hydrochloric acid Nitric acid and Ethanol). Microscopic observations were done using Nikon 600 optical microscope. Porosity of the as-sprayed samples were measured using imageJ analysis [19,20] and chemical element were identified using SEM-EDS (Phenom Desktop SEM). The heat treated samples were studied using Xray diffraction (Malvern Panalytical) with copper X-ray target with the range from 25 to 90 degree at the rate of 4 degree per minute to study the phase components present.

## Results and discussion

### Relative density

Fig 3 shows the respective elements present in the right macro concentration as a metal matrix composite. The porosity was measured to be about 2% (S1 Data) in the material which means that the density was about 98%. Visibly, aluminium distributed uniformly through the as-sprayed samples. In contrast, literature on the PM method shows the density reaches only 88%–89.3% of a mixed powder of Cu-12Al-xNi which were compacted at 550 MPa and sintered at 1000 °C [18]. Additionally, Cu-10Al-xAg also exhibit about 69%–76% relative density with compact pressure at 200 MPa and sintered at 850 °C [21]. This suggests the capability of cold spray AM to produce similar compacts density to PM but in higher density without the need for such high pressures.

It is known that traditional PM has ability to create high-density materials, typically in the range of 75% to 98% of the theoretical density, however this comes with significant limitations. The density and porosity of a product produced by the PM method is regulated by compaction pressure. The higher the compaction pressure, the greater the density, the lower the porosity, and the greater the final mechanical properties [22]. Mechanical, electrical, and other

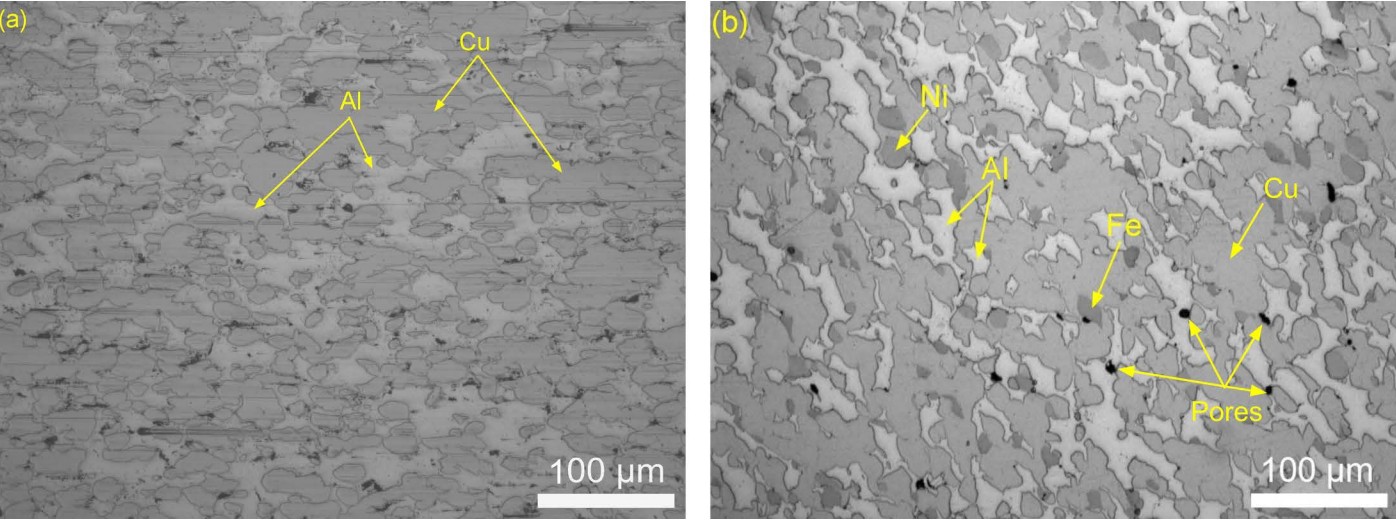

**Fig 3. Microstructure of as-sprayed alloy (a) Binary Aluminium Bronze (b) Nickel Aluminium Bronze.**

properties of a material are dependent on the density and porosity, which assumes, therefore, their indirect dependence on compaction pressure. Achieving densities above 90% typically requires very high compaction pressures (600-700 MPa) for aluminium – based material [23]. Additionally, the literature reports that iron and low-alloy steel powders require a compacted pressures at 600 MPa to 800 MPa to attain green densities of 85% to 90% of the pore-free solid density [24,25]. Copper powders compacted at 600 MPa exhibit green densities of 88-93% of the theoretical density [24]. Mahani et al. examined the impact of compaction pressure on the hardness of Cu–W-C (graphite) alloy fabricated by powder metallurgy [26]. By increasing the compaction pressure, hardness increased due to a reduction in the porosity of the material.

In comparison, cold sprayed AM and coatings have also demonstrated high green densities which do not require high pressure compact process. Aluminium and aluminium alloy coatings have been shown to possess green densities in the range of 90-98% of the theoretical density [27]. Ningsong et. al. [28] indicated that the density of pure aluminium parts produced by cold spray can reach up to 99.93%. For titanium alloy Ti6Al4V, reported densities range from 70% to 99%: Vo et. al. reported 93% density with spraying in $N_2$ [29]; Wong et. al. reported 77.6% density with spraying in air [30]. Pure copper coatings have been reported to exhibit a density of at least 98% [31]. Cold spray AM of Amstrong process Titanium powder also exhibit as high density as PM process using the same powder pressed at above 700 MPa [32,33]. Fully dense titanium parts were fabricated using the cold spray AM process. This was accomplished using low-pressure and low-temperature nitrogen as the process gas. The key to this result is attributed to the unique coral-shaped powder morphology that accelerates well in the gas flow, allowing the easy densification and deposition through cold spray [33].

In powder metallurgy, it is common practice to use blended elemental powders or a mixture of a base metal powder and master alloy powders, rather than employing pre-alloyed powders of the desired composition [34]. This approach offers several advantages: it is more cost-effective than relying completely on pre-alloyed powders, it enables tailoring of properties such as particle size distribution and flowability, it facilitates better control of interdiffusion and homogeneity during sintering compared to pre-alloyed powders, and it provides flexibility in alloy design by allowing the production of compositions that may be difficult or impossible to obtain via melting and atomization of pre-alloyed powders. With blended powders, powder metallurgy has demonstrated high densities. Our study with cold spray AM also demonstrated high densities.

## Microstructural characteristics

Phases were identified according to literature review on microstructure of NAB [35]. The microstructural evolution and phase transformations observed in aluminium bronze and nickel aluminium bronze after heat treatment are illustrated in Fig 4. Binary aluminium bronze shows the presence of both alpha phase as a major phase and beta phase (Fig 4a and b). It is evident that the materials have undergone homogenization, resulting in the formation of well-defined alloy phases. The nickel aluminium bronze exhibits a predominantly single alpha phase microstructure with fine precipitates (Fig 4c). The formation of these precipitates obtained by the aging process contributes to an increase in the material's tensile strength by means of precipitation hardening mechanisms. These microstructural observations highlight the efficacy of the heat treatment process in promoting interdiffusion, phase transformations, and the development of tailored microstructures, ultimately enhancing the mechanical properties of the respective aluminium bronze and nickel aluminium bronze alloys.

The heat treatment (HT)/sintering process is a crucial step in powder metallurgy involving mixed or blended powders, such as elemental powders or a combination of a base metal powder and master alloy powders. This thermal treatment facilitates interdiffusion and

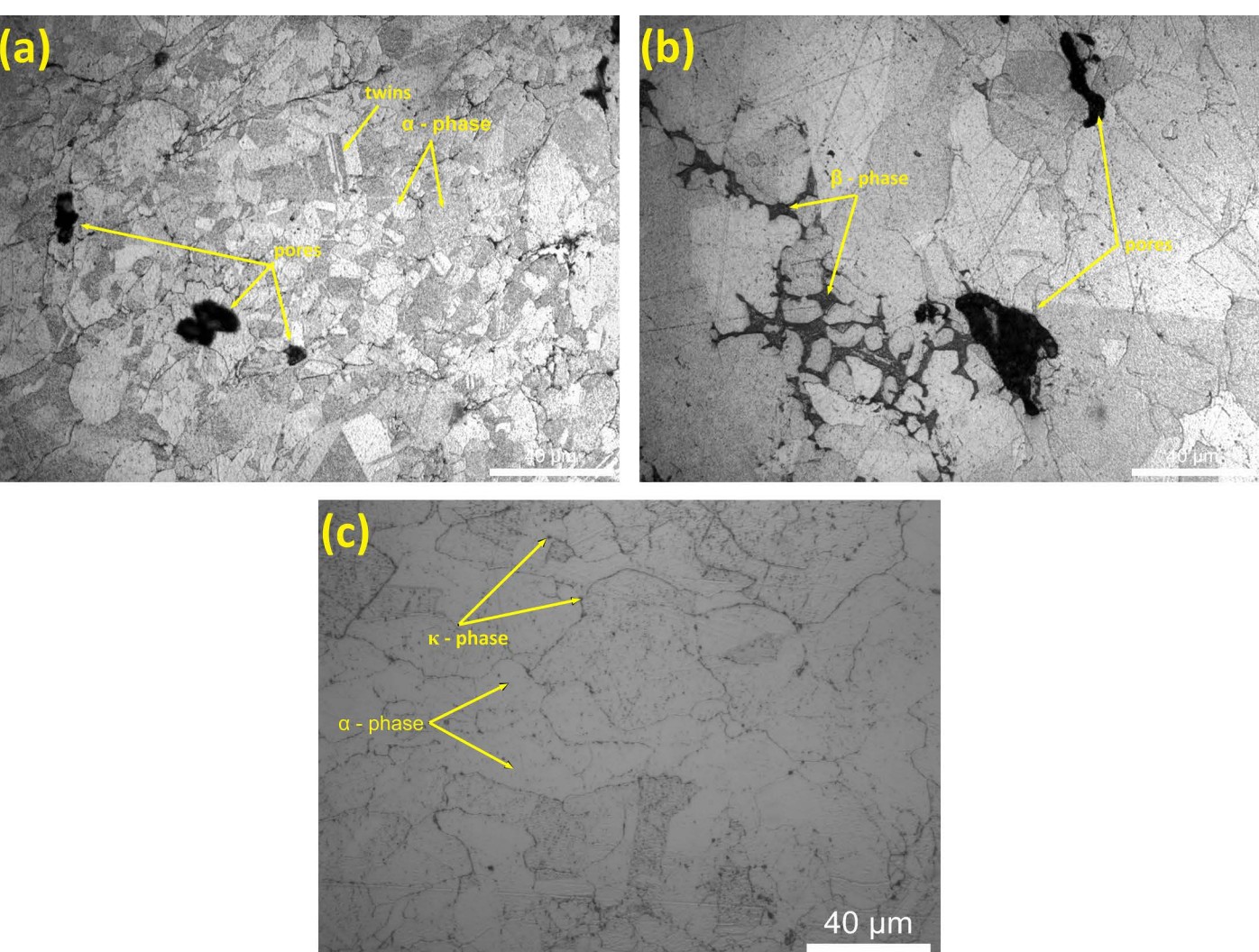

**Fig 4. After Heat treatment, microstructure of (a-b) Binary Aluminium Bronze; (c) Nickel Aluminium Bronze.**

homogenization, enabling the formation of the desired alloy composition, phase constituents, and microstructure. The extent of diffusion bonding and grain growth depends on the sintering temperature and time. Higher temperatures improve the bonding between the particles and, hence, increases the strength of a material [23,26] and densification [25]. Without proper heat treatment, the sintered components fabricated from mixed powders would exhibit incomplete alloying, high porosity, inhomogeneous microstructure, and inferior mechanical performance.

A similar phenomenon is observed in the cold spray process. The literature has demonstrated that heat treatment of cold-sprayed mixed powders, such as Cu-Ni, can help reduce the as-sprayed porosity and facilitate the formation of a homogenised alloy [36]. Literature also suggests that the strength of cold spray AM Aluminium alloy deposits can be increased by introducing small precipitates through performing heat treatment (HT) in age hardenable alloys, such as 2xxx, 6xxx, and 7xxx Al [37–39]. This is due to the fact that deformation before heat treatment is often used to promote non-uniform nucleation of precipitates in these

alloys. Other CS copper and copper alloys are also reported to have better mechanical performance after HT [40–43]. Likewise, the current study shows cold spray AM aluminium bronze and nickel aluminium bronze can form homogenized alloys through a simple heat treatment process in a low cost air atmosphere.

## Mechanical properties and further enhancing mechanical properties with the HIP process

To optimise the properties of the material, heat treatment processes were developed. The material properties post heat treatment is listed in Table 1. It should be noted that a constraint for total heat treatment time was introduced deliberately so that parts could be completely manufactured within a 24-hour cycle, including part printing, heat treatment etc. Consequently, the AB heat treatment was limited to 16.5 hours and the NAB heat treatment was limited to 18 hours. There was good evidence of the potential to improve the properties even further with longer heat treatments.

The addition of alloying elements, such as Fe and Ni, plays a crucial role in enhancing the overall properties of the materials under consideration [18]. A higher concentration of these alloying elements contributes to an increase in tensile strength; however, it is accompanied by a reduction in elongation. Interestingly, despite the cost-effective approach adopted for heat treatment, the resulting material properties approach the properties specified by ASTM B505 cast nickel aluminium bronze standard.

Furthermore, other cold-sprayed compacts were subjected to atmospheric furnace heat treatments along with Hot Isostatic Pressing (HIP), leading to the microstructures shown in Fig 5. It is evident that the inclusion of a HIP process has resulted in reduced porosity and improved homogenization within the material.

The XRD analysis, shown in Fig 6, of the NAB produced by cold spray AM revealed patterns that closely resemble those of cast NAB reported in the literature. This similarity indicates that the crystallographic structure of the phases in cold-sprayed NAB are consistent with that of traditionally cast NAB. Such congruence in the XRD patterns suggests that the cold spray AM process successfully replicates the desirable microstructural characteristics of cast NAB, thereby ensuring comparable performance and reliability in practical applications.

Table 2 suggests the mechanical properties of hipped material exceed the ASTM specification for cast Nickel aluminium bronze (C95800). The enhancement in mechanical properties achieved through the addition of alloying elements and the subsequent heat treatment process highlights the potential of this approach for tailoring the alloy characteristics of cold-sprayed materials.

Cold spray AM demonstrated mechanical properties comparable to traditional methods. For instance, the NAB produced by cold spray AM exhibited a yield strength of 289 MPa and tensile strength of 512 MPa, surpassing the typical ranges of PM methods, which often struggle to achieve similar mechanical properties due to porosity and incomplete densification. Deng et at. reported yield strength ~200–220 MPa, tensile strength ~360–400 MPa, elongation

**Table 1. Material properties with heat treatment only – no HIP.**

| | Process | Average Yield Strength (MPa) | Average Ultimate Tensile Strength (MPa) | Average Elongation (%) | Average Hardness (Brinell) |
|---|---|---|---|---|---|
| Binary Aluminium Bronze | Sintering | 160 ± 1.5 | 300 ± 2.9 | 8 ± 0.6 | 87 ± 1.7 |
| Nickel Aluminium Bronze | Sintering and aging | 257 ± 1.1 | 404 ± 2.9 | 10 ± 0.7 | 150 ± 1.9 |

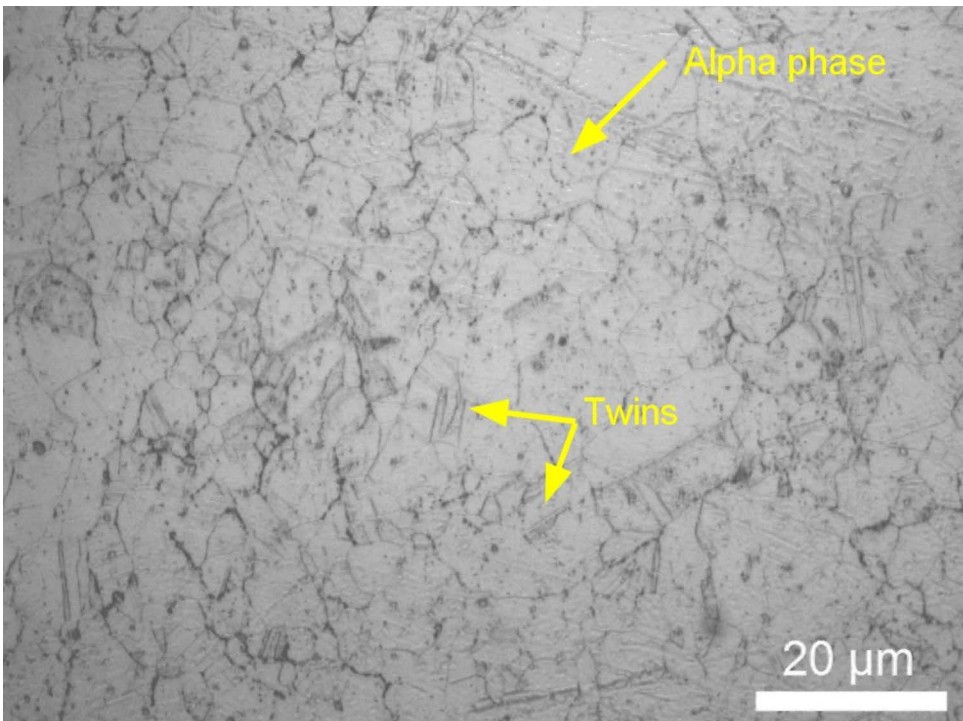

**Fig 5. Microstructure of HIPed Aluminium Bronze.**

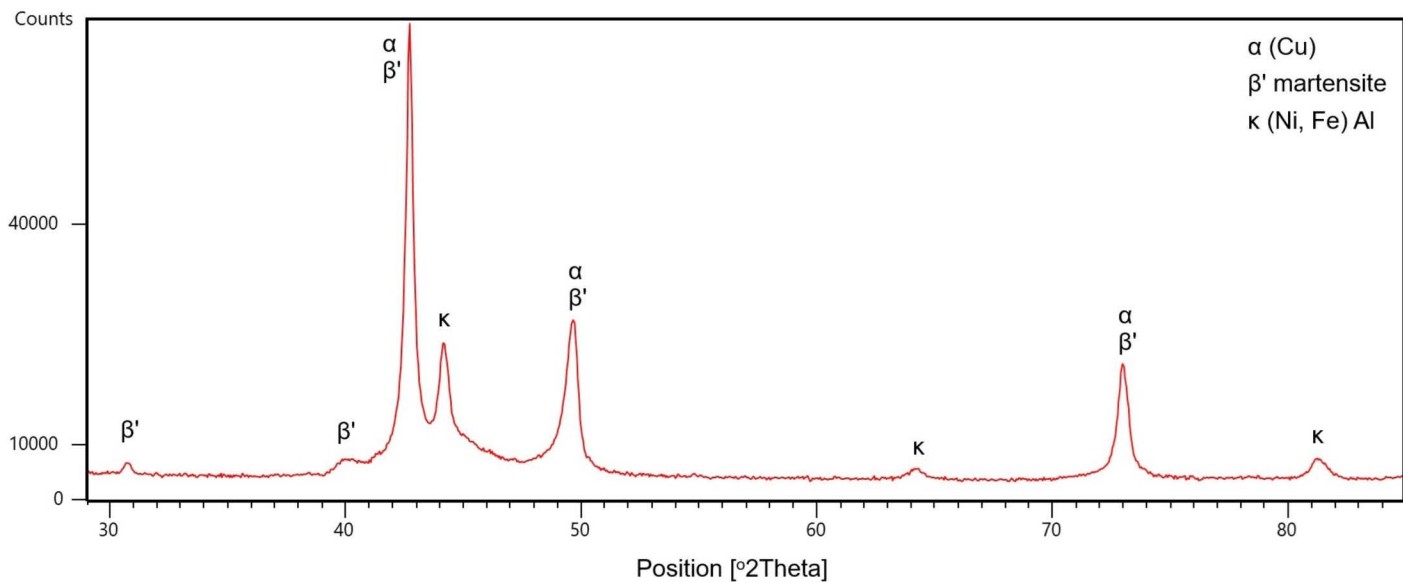

**Fig 6. XRD of HIPed Nickel Aluminium Bronze.**

~7–10%, and relative density ~89.3% for NAB processed by PM at 550 MPa compaction pressure and 1000 °C sintering [18]. Gohar et al. demonstrated NAB alloys with elongations of ~5–10% and hardness values of ~85–110 Brinell, with relative densities between 70%–85% depending on compaction pressure and sintering conditions [21]. Casting, while achieving

**Table 2. Tensile properties of NAB materials after HIP.**

|  | Process | Average Yield Strength (MPa) | Average Ultimate Tensile Strength (MPa) | Average Elongation (%) | Average Hardness (Brinell, typ) |
|---|---|---|---|---|---|
| **Nickel Aluminium Bronze** | Sintering, hipped and aging | 289 ± 1.6 | 512 ± 2.7 | 22 ± 0.3 | 160 ± 1.8 |

similar strength properties, often introduces shrinkage defects and coarse microstructures that reduce uniformity. Cold spray AM, on the other hand, achieved a refined κ phase distribution and higher density (98%) with minimal porosity. This demonstrates that cold spray AM can address some of the critical limitations of traditional PM and casting methods, offering better mechanical properties and flexibility in alloy design.

The ability to approach the properties of conventionally cast alloys while leveraging cost-effective processing techniques, or exceed them where better techniques are available, presents promising opportunities for various applications.

## Conclusions

The study shows the benefits of cold spray AM compared to traditional PM for making high-density materials, such as NAB parts. PM needs very high compaction pressures to achieve over 90% compact density. But cold spray AM can reach over 98% density with much less complexity in one off tooling and high pressure presses. This could save time and cost, making cold spray AM an interesting choice for industrial uses. The technique is also very flexible, reducing the requirement for pre-alloyed powders and presenting significant scope for tailoring alloys.

The study indicates that the microstructure achieved through cold spray AM for both materials is comparable to that of cast materials after appropriate heat treatment. In the case of aluminium bronze, the dominant alpha phase, along with a minor presence of beta phase, contributes to an overall increase in material strength. Additionally, kappa phase precipitation in NAB enhances the tensile strength of the material. The study indicates that the microstructure achieved through cold spray AM for both materials is comparable to that of cast materials after undergoing homogenization at 1000 °C for 14 hours, followed by solution treatment at 1035 °C and aging at 625 °C. These treatments facilitated the transformation of β phase into α and κ phases while refining the κ phase morphology. The κ phase evolved into a combination of $\kappa_{II}$ and $\kappa_{III}$ types, contributing to improved mechanical performance by enhancing strength without compromising ductility.

The study also shows that replacing solution treatment at HIP process at 1035 °C can further enhance the mechanical properties of cold-sprayed NAB samples, making them superior to traditionally cast NAB materials. This highlights the potential of cold spray AM to produce high-performance materials more efficiently. Thus, in general, cold spray AM offers significant advantages over PM, including improved process efficiency, material properties, and cost-effectiveness. It presents a promising method for producing complex and high-strength materials, warranting further research to optimize its parameters and expand its applications.

## Supporting information

**S1 Data. Minimal underlying dataset of the study.**
(XLSX)

## Acknowledgments

The authors would like to acknowledge the support of the technical staff at Charles Darwin University. The authors also extend their gratitude to Penn State ARL and SPEE3D for their generous support.

## Author contributions

**Conceptualization:** Steven Camilleri.

**Formal analysis:** Steven Camilleri, Tien Tran, Andrew Duguid, Kannoorpatti Narayanan Krishnan.

**Investigation:** Steven Camilleri.

**Methodology:** Tien Tran, Andrew Duguid.

**Supervision:** Kannoorpatti Narayanan Krishnan.

**Writing – original draft:** Steven Camilleri.

**Writing – review & editing:** Steven Camilleri, Tien Tran, Kannoorpatti Narayanan Krishnan.

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
