## [Decision Letter · Decision Letter 0]

8 Oct 2024

PONE-D-24-39008Cold spray as a powder metallurgy process for production of nickel aluminium bronzePLOS ONE

Dear Dr. Krishnan,

Thank you for submitting your manuscript to PLOS ONE. After careful consideration, we feel that it has merit but does not fully meet PLOS ONE’s publication criteria as it currently stands. Therefore, we invite you to submit a revised version of the manuscript that addresses the points raised during the review process.

We look forward to receiving your revised manuscript.

Kind regards,

Amitava Mukherjee, ME, Ph.D.

Academic Editor

PLOS ONE

“This research was supported (partially or fully) by the Australian Government through the Department of Education’s National Industry PhD Program (NIPhD 35017). The views expressed herein are those of the authors and are not necessarily those of the Australian Government or the Department of Education.

The work was partly funded by the generous support of Penn State ARL and SPEE3D.”

“This research was supported (partially or fully) by the Australian Government through the Department of Education’s National Industry PhD Program (NIPhD 35017). The views expressed herein are those of the authors and are not necessarily those of the Australian Government or the Department of Education.

The work was partly funded by the generous support of Penn State ARL and SPEE3D.”

4. Thank you for stating the following in the Competing Interests

“I have read the journal's policy and the authors of this manuscript have the following competing interests:

Steven Camilleri, Tien Tran and Andrew Duguid are employees of SPEE3D. This has been declared in the authorship details.”

We note that one or more of the authors have an affiliation to the commercial funders of this research study : [SPEE3D].

5. We note that your Data Availability Statement is currently as follows: [All relevant data are within the manuscript and its Supporting Information files.]

Reviewers' comments:

Reviewer's Responses to Questions

**Comments to the Author**

1. Is the manuscript technically sound, and do the data support the conclusions?

Reviewer #1: Partly

Reviewer #2: No

2. Has the statistical analysis been performed appropriately and rigorously? 

Reviewer #1: No

Reviewer #2: No

3. Have the authors made all data underlying the findings in their manuscript fully available?

Reviewer #1: Yes

Reviewer #2: Yes

4. Is the manuscript presented in an intelligible fashion and written in standard English?

Reviewer #1: No

Reviewer #2: No

5. Review Comments to the Author

Reviewer #1: Fisrst of all, the paper is quite attractive since it touchs a very ineresting topic, in particular related to this complexes systems, NAB. It should be modify in order to point out the relevance of microstrcuture and the relation with the process per se and the impact on mechanical properties. The microstrcutural analysis is poor. Figures 3 and 4 needs to identify better the phases and microprepitates. So, have K phases this new cold spray AM new NAB?. Figure 3, how did you identify this "beta phase"?. It looks like rather defects.

Where is the Cu in Future 2?. what are this black spot of Figure 2 b?. Fig 3 and Figure 4 perhaps need more time for the etching in order to reveal better the microstructure, since it is the ley point of your research.

You mention "better properties " of your cold sprayed AM NABm, in comparison traditional methods. It should be interesting point out the relevance of phases, percentages, average of the grain size, distribution and so on.

The conclusions need to be reviewed in order to point out the real effect of microstructure.

The references have some gramatical mistakes ( ca ref 1).

Reviewer #2: The authors present a study on cold spray additive manufacturing of nickel aluminium bronze, and compared the tensile property under both HIPped and heat treated conditions. Please find my comments in the attached pdf.

6. PLOS authors have the option to publish the peer review history of their article (what does this mean? ). If published, this will include your full peer review and any attached files.

**Do you want your identity to be public for this peer review?** For information about this choice, including consent withdrawal, please see our Privacy Policy .

Reviewer #1: No

Reviewer #2: No

---

## [Author Response · Author response to Decision Letter 1]

18 Nov 2024

Thank you for your constructive feedback on the manuscript titled " Cold spray as a powder metallurgy process for production of nickel aluminium bronze." Your insightful comments have significantly improved the manuscript, and I am grateful for your guidance.

Address to reviewer’s comments

Reviewer #1:

First of all, the paper is quite attractive since it touchs a very ineresting topic, in particular related to this complexes systems, NAB. It should be modify in order to point out the relevance of microstrcuture and the relation with the process per se and the impact on mechanical properties.

The microstrcutural analysis is poor.

• We have added more information on microstructure analysis into the manuscript.

Figures 3 and 4 needs to identify better the phases and microprepitates. So, have K phases this new cold spray AM new NAB?. Figure 3, how did you identify this "beta phase"?. It looks like rather defects.

• All phases identified in the etched microstructure as example from the literature.

Where is the Cu in Future 2?. what are this black spot of Figure 2 b?.

• We have added Cu arrow into the figures 2b to highlight the location of Cu. The black spots are all porosity. We also added this information into the figure.

Fig 3 and Figure 4 perhaps need more time for the etching in order to reveal better the microstructure, since it is the ley point of your research.

• We have etched the samples again and replaced the old images.

You mention "better properties " of your cold sprayed AM NABm, in comparison traditional methods. It should be interesting point out the relevance of phases, percentages, average of the grain size, distribution and so on.

• Thank you for your valuable suggestions. Due to limited facilities, we are unable to precisely measure phase quantities and grain size. The term "better properties" in the manuscript refers primarily to the material's properties. We have revised this section in the manuscript to clarify and prevent any confusion.

The conclusions need to be reviewed in order to point out the real effect of microstructure.

• We have added more information in the conclusion section.

The references have some gramatical mistakes ( ca ref 1).

• We have made correction.

Reviewer #2:

1. But cold spray also induces significant compressive force during deposition?

• Yes. It is noted.

2. AM?

• AM stands for Additive Manufacturing. We have made change in the manuscript.

3. “The results indicate that the cold spray process, combined with appropriate heat treatments, can produce NAB alloys with desirable microstructures and mechanical properties comparable to those achieved by traditional cast methods, and superior to PM methods.” – Examiner mentions, “This is a generic statement and does not provide any key information about the actual results of this study in both qualitative and quantitative values.”

• We have made change in the manuscript as follows:

“The results indicate that the cold spray process, combined with appropriate heat treatments, can produce NAB alloys with desirable microstructures containing fine κ phases and mechanical properties with above 280 MPa yield strength, 500 MPa tensile strength and 20% elongation which are comparable to those achieved by traditional cast methods, and superior to PM methods.”

4. Capitalised Cold Spray.

• We have made change in the manuscript.

5. “Nickel aluminium bronzes (NAB) are a popular set of alloys due to their excellent combination of properties. They have high strength and toughness, high wear, galling and cavitation resistances, are non-sparking, good corrosion and stress corrosion properties and biofouling resistance [1]. They are used extensively in aircraft landing gear bearings, marine propellers, pumps and valves, gears, and in non-sparking tools. They can be produced in cast, wrought or forged forms as well as by powder metallurgy. There are many different compositions of NAB alloy, but they generally contain up to 12% aluminium, 5% nickel, 7% iron, 3% manganese, and 2% silicon. The guide by Copper Development Association [1] is an excellent source of information on NAB. Aluminium is the primary element that strengthens the NAB alloys. Nickel and iron are added to improve corrosion and erosion resistance. Addition of iron also refines the structure and improves toughness. Manganese is added to improve fluidity of the alloys when casting. When nickel and iron, about 5%, are added to copper-aluminium alloy, two phases α and κ are produced. When NAB is quenched from 900oC, both α phase, martensite β and κ phases can form. On tempering at 500-715oC, the martensite phase transforms to α (Cu-rich) and eutectoid phase (κ and γ2) and secondary κ phases. The κ phase is found in 4 morphologies, locations and distributions. They are: ΚI – rosette form and iron-rich, κII – spheroidised form at”. Examiner – “This information is primarily summarized from one source [1]. Please provide additional information from other published literature that can support these statements.”

• We have added more information along with references in the manuscript.

6. “Various kappa phases containing Ni and Fe are known to increase the mechanical properties of NAB. A decrease in elongation has been attributed to excess κ phases which form with an increase in aluminium content above nominal.” Examiner – “Please provide reference for this claim. Also mention the nominal aluminium content.”

• We have added more information along with references in the manuscript.

7. “WAAM” Examiner – “WAAM is mentioned in the previous paragraph as wire arc additive manufacturing.”

• We have made change in the manuscript.

8. “The first several paragraphs of the Introduction section was heavily focussed on microstructural phases exhibited in NAB alloys when subjected to different processing conditions. However, the later part of this section discussed more on additive manufacturing of NAB and the issues when printing using pre-alloyed powders. These two sections are quite distinct and there is no correlation drawn between microstructural variations in cast/forged/pht NAB parts with those of AMed NAB, as well as the mechanical property comparisions, which are the key results studied in this research work. Moreover, the Introduction lacks a pragraph on the research gaps and the importance/novelty of this study.

In my opinion, this section has to be considerably improved to better present the state-of-the-art literature on CSAM, its application to not only NAB but other related Cu-alloys.”

• We have made changes in the manuscript.

9. Why was Aluminium bronze studied in this work? The rationale behind this alloy addition should be discussed in the Introduction section.

• Aluminium bronze was selected to take advantages of using elemental blended with widely different melting temperatures.

10. What are the chemical compositions of the powders, their morphology, and their PSD? Was this the AL6061 alloy? What was the chemical composition of this powder? Is it 9.9% by weight or volume?

• Thank you for your insightful questions. We have revised the manuscript to include additional details about the powder, including its chemical composition. However, please note that the powder particle size distribution (PSD) is proprietary information, so we are unable to disclose it in the manuscript.

11. How long was the mixing carried out? Was it at room temperature? More experimental details are to be provided.

• We have made changes in the manuscript.

12. What was the powder feed rate?

• It is 60 g/min. We have made changes in the manuscript.

13. Were the same process parameters used to print both aluminium bronze and nickel aluminium bronze coupons?

• Yes. The two materials were deposited with the same parameters.

14. Does normal properties in this instance refers to cast or wrought mechanical properties?

• It refers to cast mechanical properties. We have changed the word normal to cast mechanical properties.

15. Why were these heat treatment temperatures selected? Can this heat treatment protocol be shown in the temperature schedule graph with respective heating and cooling rates? Also, was this HT followed for both AB and NAB?

• We have added the HT for the AB and a graph in the manuscript. Additionally, we have added an explanation for the temperature chosen for heat treatment.

16. Please present a table to show how many conditions were studied and mechanically tested in this work?

• Tables 1 and 2 provided the details.

17. For the XRD analysis, what was the x-ray source used? Also, the angular data capture rate?

• We have made changes in the manuscript as below:

The heat treated samples were studied using Xray diffraction (Malvern Panalytical) with copper X-ray target with the range from 25 to 90 degree at the rate of 4 degree per minute to study the phase components present.

18. There is no discussion of actual observations drawn from the micrographs shown in Figure 2.

• We have made changes in the manuscript.

19. How was density measured? This is not discussed in the Methodology section.

• The density was measured by 100% - %porosity. We have added this in the methodology section.

20. Does this statement refer to similar composition or similar properties?

• The statement refers to similar density to PM. We have made changes to the statement to make it clearer.

21. This section is basically a discussion of published literature to compare relative densities achieved with that of PM process for various materials compared to that of cold spray process, which is relevant for an Introduction section. In this section, please discuss as to why the relative density of your NAB coupons in the as-sprayed condition is significantly higher than that reported in literature for other Cu-based cold sprayed parts. Is it owing to the nozzle design or process parameters or powder characteristics. A theory should be proposed and validated with subsequent experimentation. A recent study showed that the porosity percentage increases in heat treated aluminium bronze parts when compared with as-sprayed condition. https://doi.org/10.1016/j.nxmate.2024.100312

• It is still unclear as to why we got high density for the materials. It could be attributed to the nozzle design, process parameters and powder characteristics. We have not use any other machine so we are unable to comment on comparing the density with others.

22. Why? Is it only for AB or NAB parts? More discussion of results is required in this section.

• This statement refers to both materials.

23. The first two paragraphs of this section should be discussed after presenting the results of this study. So, Fig. 3 should be presented and the key observations discussed, followed by literature based discussions.

• We have made changes to the manuscript.

24. How were the phases identified and correlated in the microstructure? Is there any EDS carried out to validate the statements made here.

• The phases were identified by comparing with phases from literature.

25. Since 5 coupons were tensile tested at each condition, the standard deviation should be mentioned for all mean values shown in Table 1.

• We have added this information.

26. Why was only sintering carried out for AB? Why not aging treatment?

• There is no aging behaviour in AB system with 9.9%Al, thus no aging treatment was needed.

27. These properties should also be presented in Table 1 along with the mechanical properties of similar parts produced by CSAM for better comparision. Table 2 should be merged with Table 1 for better comparision.

• These properties in different section, thus it is unsuitable to move the table to the previous section.

28. The Abstract mentions that the mechanical properties of cold sprayed NAB parts outperformed those of cast and PM parts. Where is this information presented in the Results and Discussion sections?

• The statement was in section 3.3 as follows:

“Table 2 suggests the mechanical properties of hipped material exceed the ASTM specification for cast Nickel aluminium bronze (C95800).”

---

## [Decision Letter · Decision Letter 1]

1 Jan 2025

PONE-D-24-39008R1Cold spray as a powder metallurgy process for production of nickel aluminium bronzePLOS ONE

Dear Dr. Krishnan,

Thank you for submitting your manuscript to PLOS ONE. After careful consideration, we feel that it has merit but does not fully meet PLOS ONE’s publication criteria as it currently stands. Therefore, we invite you to submit a revised version of the manuscript that addresses the points raised during the review process.

We look forward to receiving your revised manuscript.

Kind regards,

Amitava Mukherjee, ME, Ph.D.

Academic Editor

PLOS ONE

Journal Requirements:

Reviewers' comments:

Reviewer's Responses to Questions

**Comments to the Author**

1. If the authors have adequately addressed your comments raised in a previous round of review and you feel that this manuscript is now acceptable for publication, you may indicate that here to bypass the “Comments to the Author” section, enter your conflict of interest statement in the “Confidential to Editor” section, and submit your "Accept" recommendation.

Reviewer #1: All comments have been addressed

Reviewer #2: All comments have been addressed

2. Is the manuscript technically sound, and do the data support the conclusions?

Reviewer #1: Partly

Reviewer #2: Yes

3. Has the statistical analysis been performed appropriately and rigorously? 

Reviewer #1: N/A

Reviewer #2: Yes

4. Have the authors made all data underlying the findings in their manuscript fully available?

Reviewer #1: Yes

Reviewer #2: Yes

5. Is the manuscript presented in an intelligible fashion and written in standard English?

Reviewer #1: Yes

Reviewer #2: Yes

6. Review Comments to the Author

Reviewer #1: It should be mandatory specify in conclusion the heat treatments that you argued in the way "The study indicates that the microstructure achieved through cold spray AM for both materials is comparable to that of cast materials after appropriate heat treatment. ". Which heat treatment?. in which way is altering the microstructure?. does affected to beta/alphase percentages phases? K distribution?percentajes?shapes? and so on.

Are you sure that you can identify the chemical elements defined in Figure 3?

You conclude that "This highlights the potential of cCold sSpray AM to produce high-performance materials more efficiently. ". Do you consider that defects ( pores, shrinkages, etc) can penalize the performance of this technique?

Reviewer #2: The authors have addressed most of the previous comments. I have few more comments in the attached pdf.

7. PLOS authors have the option to publish the peer review history of their article (what does this mean? ). If published, this will include your full peer review and any attached files.

**Do you want your identity to be public for this peer review?** For information about this choice, including consent withdrawal, please see our Privacy Policy .

Reviewer #1: No

Reviewer #2: No

---

## [Author Response · Author response to Decision Letter 2]

28 Jan 2025

Thank you for your constructive feedback on the manuscript titled " Cold spray as a powder metallurgy process for production of nickel aluminium bronze." Your insightful comments have significantly improved the manuscript, and I am grateful for your guidance.

Address to reviewer’s comments

Reviewer #1:

It should be mandatory specify in conclusion the heat treatments that you argued in the way "The study indicates that the microstructure achieved through cold spray AM for both materials is comparable to that of cast materials after appropriate heat treatment. ". Which heat treatment?. in which way is altering the microstructure?. does affected to beta/alphase percentages phases? K distribution ? percentajes ? shapes ? and so on.

Answer: Thank you for your insightful comments. We agree that the heat treatment details need to be explicitly stated in the conclusion. We will modify the conclusion to specify the heat treatments applied and their impact on microstructure, particularly regarding the α/β phase distribution and κ phase morphology.

Are you sure that you can identify the chemical elements defined in Figure 3?

Answer: Yes. We acknowledge the need to clarify the methodology used for elemental identification in Figure 3. The elements were identified through EDS, which was conducted on selected areas of the sample. To ensure accuracy, we will explicitly mention this method in the manuscript.

You conclude that "This highlights the potential of cCold sSpray AM to produce high-performance materials more efficiently. ". Do you consider that defects ( pores, shrinkages, etc) can penalize the performance of this technique?

Answer: Yes, defects such as porosity and incomplete bonding can influence the mechanical performance of cold spray AM materials. However, our study demonstrates that post-processing techniques, including heat treatment and hot isostatic pressing (HIP), significantly reduce porosity and improve mechanical properties. We will add a discussion on how these defects can be mitigated.

Reviewer #2:

1. Another useful publication to consider for this section: https://doi.org/10.1016/j.nxmate.2024.100312

Answer: We have added this into the manuscript. Thank you for the suggestion.

2. The abstract mentions that the study aims to compare NAB manufactured by cold spray AM to those produced by traditional methods (casting and PM), but the discussion lacks critical comparison.

Answer: Thank you for pointing this out. We have discussed in above sections, however we agree that is not enough. A further detailed comparison of the mechanical properties between cold spray AM and traditional methods were added to the discussion section. This will highlight the advantages and limitations of cold spray AM.

3. The κ phase is mentioned in the abstract, but no quantitative evidence (e.g., XRD or EBSD) is provided to verify this feature.

Answer: Thank you for your observation. The presence of κ phases was inferred from the XRD pattern (Fig. 6), where characteristic peaks corresponding to κ phases were observed. Although quantitative analysis of phase fractions was not performed, the microstructural refinement visible in optical microscopy and the precipitation of κ phases following heat treatment align with findings reported in literature. Future work should employ EBSD to confirm and quantify the distribution and orientation of κ phases to further validate these observations.

---

## [Editor Report · Decision Letter 2]

31 Jan 2025

Cold spray as a powder metallurgy process for production of nickel aluminium bronze

PONE-D-24-39008R2

Dear Dr. Krishnan,

We’re pleased to inform you that your manuscript has been judged scientifically suitable for publication and will be formally accepted for publication once it meets all outstanding technical requirements.

Kind regards,

Amitava Mukherjee, ME, Ph.D.

Academic Editor

PLOS ONE
---

## [Editor Report · Acceptance letter]

PONE-D-24-39008R2

PLOS ONE

Dear Dr. Kannoorpatti,

I'm pleased to inform you that your manuscript has been deemed suitable for publication in PLOS ONE. Congratulations! Your manuscript is now being handed over to our production team.

Kind regards,

on behalf of

Professor Dr. Amitava Mukherjee

Academic Editor

PLOS ONE